# Research on the Impact of Digital Literacy on Farmer Households’ Green Cooking Energy Consumption: Evidence from Rural China

**DOI:** 10.3390/ijerph192013464

**Published:** 2022-10-18

**Authors:** Lei Zhao, Yongqi Zhang, Haixia Zhang

**Affiliations:** College of Economics, Sichuan Agricultural University, Chengdu 611130, China

**Keywords:** digital literacy, green cooking energy, non-agricultural employment, information acquisition, CMP model

## Abstract

In the era of the digital economy, farmers’ digital literacy has a profound impact on household green cooking energy consumption. Based on data from the China Family Panel Studies (CFPS) in 2018, this paper constructs a digital literacy index using an entropy method and employs the Probit model regression and mediation effect model test to study the impact effect and theoretical mechanism of individual digital literacy on household green cooking energy consumption. The research results show that the improvement of digital literacy can effectively promote the consumption of green cooking energy in households. After using the IV-Probit model and CMP model to solve the endogeneity, this conclusion remains valid; The heterogeneity analysis shows that the impact of digital literacy on green cooking energy consumption of households in different regions and different income stages is different. This performance is specific to the eastern and western regions and low-income households. The improvement in digital literacy can significantly promote green cooking energy consumption in rural households; however, in the central region and high-income households the improvements were insignificant. Mechanism analysis shows that digital literacy has a significant positive impact on household green cooking energy consumption through non-agricultural employment and information acquisition. Based on this, it is suggested that the construction of a digital countryside should not only consider the construction of digital infrastructure, but also reasonably guide the cultivation of the internal digital literacy of the construction subject. Moreover, the cultivation of digital literacy should not only focus on regional differences, but also focus on key subjects and implement precise cultivation. We should give full play to the synergistic effect of digital literacy, and pay attention to the non-agricultural employment of farmers and information elements.

## 1. Introduction

The International Energy Agency pointed out in “Energy Balances of OECD Countries” that household energy consumption has been increasing in recent years and has become the main source of global energy demand and carbon emission growth [1]. Meanwhile, for the first time since 2011, the growth rate of energy consumption in the household sector has exceeded that in the industrial sector, with sustained growth [2]. Although the rapid development of China’s urbanization has caused a large number of rural populations to flood into cities and towns, the total carbon emission from rural domestic energy consumption has maintained rapid growth. According to data from China’s statistical yearbook, the total per capita domestic energy consumption in China’s rural areas has been on the rise since 2001, reaching the same level as the total per capita domestic energy consumption in cities and towns in 2017, and surpassing the total per capita domestic energy consumption in cities and towns in 2019, indicating that the current energy consumption in China’s rural areas tends to be “quantitative” rather than “qualitative”. The data in the “Main Data Bulletin of the Third National Agricultural Census” show that 101.77 million and 55.06 million rural residents in China use firewood and coal as their main energy sources for cooking and heating, accounting for 44.2% and 23.9%, respectively. Traditional biomass energy is still the mainstream trend in China’s rural domestic energy consumption, and the rural energy consumption structure is unbalanced. Both the 19th National Congress Report and the 14th Five-Year Plan explicitly proposed to promote the revolution in energy production and consumption and to build a modern energy system, indicating that China’s rural energy consumption structure will be transformed into low-carbon consumption. However, the issue of rural energy consumption has not been paid attention to by the government and relevant departments for a long time [3], and rural household energy consumption is diversified, independent and scattered, which results in the government being unable to regulate rural energy consumption through effective regulatory policies [4,5]. The selective neglect of rural energy consumption has caused many negative impacts on the regional ecological environment and the health status of farmers [6,7]. Therefore, guiding the green transformation of household cooking energy is beneficial to promote the low-carbon transformation of rural energy consumption, which is of great significance to the construction of a beautiful countryside and the realization of rural revitalization.

The influencing factors of farmers’ green energy consumption mainly focus on education, gender, age, income, etc. In terms of farmers’ education level, Sun et al. [8] pointed out in their empirical study that the education level of the head of household has a positive impact on the use of biogas energy, and the higher the education level, the cleaner the energy such as biogas may be used. In a study on the determinants of household energy use in Bhutan, Das et al. [9] pointed out that education tends to rely more on green energy regarding energy choices for lighting, cooking and heating. In terms of gender, Qu et al. [10] pointed out in their research on biogas energy that the gender of the head of household plays a certain role in the construction of biogas digesters and the decision to use biogas. As the main users of domestic energy in rural households, women are more willing to use efficient and convenient cooking energy [11]. Research by Farhar et al. [12] and Das et al. [9] found that when the head of household is a female, the household energy choices tend to be clean energy, especially in the use of lighting and cooking energy, green energy such as electricity is the first choice. Farsi et al. [13] employed the ordered discrete choice model to study and analyze the energy consumption choice behavior of urban households in India, and concluded that gender is an important factor affecting energy consumption. However, some studies have come to the opposite conclusion. Tabuti et al. [14] found that the proportion of women is positively correlated with the consumption of firewood by studying the influence of family members’ structure on energy use. In terms of age structure, Liu et al. [15] found that young people are more receptive to new technologies and new things, and thus young-headed households are more willing to adopt and use green energy; Twumasi et al. [16] pointed out in their study of credit availability on green energy consumption of rural households that age has an inverted U-shaped effect on the use of green energy, i.e., the preference for clean energy changes from partiality to inhibition with the growth of age. In terms of income, Tian and Chang’s research on household energy in the Beijing–Tianjin–Hebei region showed that with an increase in income, the use of firewood and coal will continue to decline, while the use of solar energy will increase [17]. There were also studies investigating the impact on farmers’ green energy use from the perspectives of non-farm employment [18], happiness [19], ecological public welfare [20] and internet use [21]. Many scholars had provided theoretical reference for the transformation of rural energy consumption in China through different research perspectives.

At the same time, the Opinions of the CPC Central Committee and the State Council on Promoting the Key Work of Rural Revitalization in an All-Round Way in 2022 pointed out for the first time that efforts should be made to promote the construction of digital villages and to strengthen the training of farmers’ digital literacy and skills. The Action Plan for Digital Village Development (2022–2025) also proposes to strengthen the combination of digital construction and green village building. Therefore, under the background of digital construction rising to the national top-level design, the rapid development of digital information technology will have a far-reaching impact on the decision-making of individual farmers. Digital literacy, as the basic literacy for farmers to effectively use digital tools, access digital resources, and promote information sharing [22], plays an important role in promoting the transformation of rural energy consumption. Digital literacy was first proposed by Gilster in 1997 [23]. It is defined as “the ability to understand the digital resources displayed by digital media such as computers and the true meaning contained in information”. It emphasizes the ability to transmit and process information in the digital age, and the key is the ability to integrate digital information. Subsequently, Eshet-Alkalai [24] constructed a more specific measurement framework of digital literacy according to the fuzzy definition of digital literacy. It divided digital literacy into five dimensions, namely “picture-visual literacy”, “recreation literacy”, “classified thinking literacy”, “information literacy” and “social-emotional literacy”. It emphasized residents’ ability to understand, create, separate, criticize and communicate digital information. Based on the deepening of research on digital literacy and the expansion of its influence, UNESCO issued the Global Framework for Digital Literacy in 2018, which aims to improve citizens’ digital literacy and monitor the quality of digital literacy education in various countries around the world. It further defines digital literacy by constructing 7 literacy areas and 26 specific indicators. Nedungadi et al. [25] based on the perspective of differences between urban and rural areas, proposed that the digital literacy framework for rural residents should contain six dimensions, namely, information, health, finance, digital government, digital security, and online education. Su and Peng [26] comprehensively measured the composition and status quo of China’s rural residents’ digital literacy based on the four dimensions of digital general literacy, digital social literacy, digital creative literacy, and digital security literacy and 12 specific indicators. Shan et al. [27] established an evaluation index system for the development of China’s national digital literacy and skills based on the four scenarios of digital life, digital learning, digital work, and digital innovation proposed in the Action Plan for Promoting National Digital Literacy and Skills. Overall, with the continuous development of digital technology, digital literacy, as an applied digital human capital, represents an important way to expand the dividend of population quality, and also an important starting point for realizing rural revitalization, which bears great significance in promoting the green and sustainable development of rural areas.

Although some studies have discussed the impact of the application of digital technology on farmers’ green energy consumption [21], further improvements are still needed. In the rapid development of digital economy, the application of digital technologies such as the Internet has had a profound impact on farmers’ daily lives. The improvement in digital literacy is not only a simple application of digital technologies but also a deep application closely intertwined with production and life. However, in the current research on digital literacy and clean energy consumption, China demonstrates a relative lack of evidence. At the same time, the research on rural energy is mostly based on macro-data such as carbon emissions [28,29], and the analysis on green cooking energy of farmers’ households is relatively scarce.

Based on the above analysis, this paper uses data from the China Family Panel Studies (CFPS) in 2018 as a sample. Through the construction of farmers’ digital literacy indicators, this work analyzes whether digital literacy affects farmers’ household green cooking energy consumption from a multi-dimensional perspective, explores its internal mechanism to observe whether the further development of digital literacy can change China’s rural energy consumption structure and provide a new opportunity for the construction of a beautiful countryside, improvement of the rural living environment and the realization of rural revitalization. The structure of this paper is as follows: Section 2 provides the theoretical analysis and research hypothesis; Section 3 presents the data source and model sets; empirical analysis is provided in Section 4; conclusion and suggestions are presented in Section 5.

## 2. Theoretical Analysis and Hypothesis

With the rapid development of the digital internet, the digital literacy of individual farmers is an important factor that affects family decision-making. In theory, digital literacy, as human capital in the digital era, is mainly embodied in the use, processing, and creation of new value of digital technology, which in turn can influence individual decision-making by changing the relative income and cost of farmers’ participation in various economic activities [30]. As “technical” human capital, digital literacy can enable the integration of individual farmers into the digital society and their enjoyment of digital welfare, providing internal support for non-agricultural employment. By improving farmers’ digital literacy, farmers can improve their family productivity and information acquisition ability at a lower cost, thus promoting their families’ consumption of green cooking energy. In general, digital literacy affects the energy consumption of green cooking in households through the following two channels:

First, digital literacy improves farmers’ awareness and use of digital technology, which can influence farmers’ household green cooking energy consumption by promoting non-agricultural employment. Previous studies have shown that non-agricultural employment can promote the transformation of household energy into green sustainable energy [31]. Specifically, non-agricultural employment takes up the time for farmers to participate in agricultural operations. Consequently, this limits the scale of farmers’ land operations and results in the outflow of farmland and forest land, thus reducing the availability of solid fuels such as firewood and straw and increasing the use of clean energy to meet the needs of household energy [32,33,34]. On the other hand, the non-agricultural transformation of farmers’ livelihoods has accelerated the process of urban–rural integration and continuously improved the construction of energy infrastructure. Farmers will give priority to energy efficiency, convenience, and cleanliness [35,36]. The promotion of digital literacy in non-agricultural employment is reflected in the following aspects: First, the “technical” features of digital literacy can effectively enhance the use of digital tools such as the Internet by farmers, and can improve the working efficiency of farmers and their competitiveness in the labor market, thus increasing the probability of employment and entrepreneurship [30,37]; Second, digital literacy can effectively improve farmers’ information processing ability and provide more opportunities for non-agricultural jobs, such as e-commerce, e-commerce live broadcast and other new jobs with higher digital requirements; Third, digital literacy can affect farmers’ ability to participate in and profit from financial markets [27], effectively ease financing constraints and give full play to the supporting role of financial capital in non-agricultural employment [38], such as access to more financial products and high-quality financial services. To sum up, the improvement of farmers’ digital literacy can effectively increase the probability of non-agricultural employment, and thus promote the consumption of green cooking energy in households.

Furthermore, digital literacy, as a kind of digital ability, can improve an individual’s ability to obtain information and reduce information asymmetry, thus affecting the household’s green cooking energy consumption. The enhancement of digital literacy can effectively improve farmers’ environmental behavior attitude and health risk awareness. This may be achieved through the use of digital tools such as mobile phones and computers that can enable farmers to more intuitively recognize damage to the ecological environment caused by the use of traditional energy such as straw burning. Digital methods such as browsing pictures, watching videos, and exposure to related publicity concerning the impact of traditional energy use on the agricultural ecological environment system [39], strengthen environmental risk awareness, and thus enhance the attention to ecological environmental protection [40]. It will also allow farmers to realize that the damage of ecological environment protection is closely related to the health of individuals, to strengthen health risk awareness [41] and promote the consumption of green cooking energy by farmers’ families.

To sum up, this paper puts forward the following three hypotheses:

**Hypothesis** **1.**
*The improvement of farmers’ digital literacy will promote household green cooking energy consumption.*


**Hypothesis** **2.**
*Farmers’ digital literacy promotes household green cooking energy consumption by promoting non-agricultural employment.*


**Hypothesis** **3.**
*Farmers’ digital literacy promotes household green cooking energy consumption by broadening access to information.*


## 3. Sample Description and Model Construction

### 3.1. Data Source

The research data used in this paper are from China Family Panel Studies (CFPS) in 2018. CFPS is implemented by the China Social Sciences Research Center (ISS) of Peking University. It aims to track and collect data on individual, family, and community levels to reflect the changes in China’s society, economy, population, education, and health. The sample includes 25 provinces (autonomous regions and municipalities) in China excluding Hong Kong, Macao, and Taiwan Province, as well as Xinjiang Uygur Autonomous Region, Tibet Autonomous Region, Qinghai Province, Inner Mongolia Autonomous Region, Ningxia Hui Autonomous Region, and Hainan Province. It is comprehensive and representative. To effectively reveal the impact of digital literacy on the green cooking energy of farmers’ households, 4100 data files were finally obtained after merging and filtering the individual, family, and community data. The specific screening process is as follows: firstly, as the research group is the group that can have certain decision-making power over the household cooking energy consumption, the head of the household sample is used when selecting the sample, and the age is controlled to be between 16 and 65 years old; secondly, after removing the urban residents, the missing values and abnormal values in the core variable cooking fuel are removed. Finally, the values of the remaining control variables are further cleaned.

### 3.2. Variable Selection

#### 3.2.1. Explained Variables

The explained variable is green cooking energy. Drawing on the research of He et al. [21] and Liu et al. [42], the combustion of firewood and coal produces a large amount of carbon dioxide and contributes to air pollution. Therefore, this study is measured according to the question “which fuel is the most important in your family for cooking?” in the CFPS questionnaire, the answer options are 1–6 representing firewood, coal, liquefied gas, natural gas, solar energy, and electricity, respectively. The answer options 1 and 2 are defined as non-green energy for household cooking, with a value of 0; Answers 3–6 define household cooking energy as green energy, with a value of 1. As can be seen from Table 1, on the whole, the proportion of farmers choosing green cooking energy is 56.9%, and the proportion choosing non-green cooking energy is 43.1%. The cleaning level of cooking energy is far lower than the national average.

#### 3.2.2. Explanatory Variables

The explanatory variable is digital literacy. Based on the above analysis, this paper adopts the definition of digital literacy given by Eshet-Alkalai [24] and the realization scenario of digital literacy [27] for reference, and comprehensively measures the level of digital literacy of farmers according to the five questions in the questionnaire: frequency of using the Internet to learn (times), frequency of using the Internet to work (times), frequency of using the Internet to socialize (times), frequency of using the Internet to entertain (times) and frequency of using the Internet for business activities (times). In the specific measurement process, the entropy method is used for measurement and analysis. The specific calculation process is as follows.

First, to eliminate the differences caused by different dimensions of the original data, the range standardization method is used to process the original data, and since all indicators are positive indicators, the following algorithm is selected for data standardization:(1)$$Sd_{ij}=\left(R_{ij}-R_{jmin}\right)/\left(R_{jmax}-R_{jmin}\right)$$

Second, the weights of each indicator are calculated as follows, by standardizing the value $Sd_{ij}$ and calculating the weight $P_{ij}$ of the value of the i farmer’s indicator under the j indicator:(2)$$P_{ij}=\frac{Sd_{ij}}{\sum_{i=1}^{n}Sd_{ij}}$$

Third, the entropy value $E_{j}$ of the j indicator is measured:(3)$$E_{j}=-k\sum_{i=1}^{n}P_{ij}\ln\left(P_{ij}\right);k=1/\ln\left(n\right)$$

Then, calculate the information entropy redundancy $D_{j}$ for the j indicator:(4)$$D_{j}=1-E_{j}$$

Further, the weight $W_{j}$ of the entropy value of the j indicator is calculated:(5)$$W_{j}=D_{j}/\sum_{i=1}^{n}D_{j}$$

Finally, the comprehensive digital literacy score $S_{i}$ of the i farmer is calculated:(6)$$S_{i}=\sum_{j=1}^{m}W_{j}\times Sd_{ij}$$

#### 3.2.3. Control Variable

To better demonstrate the relationship between digital literacy and farmer household green cooking energy consumption, we draw on previous relevant research and start from individual characteristics, family characteristics, and village characteristics to prevent the regression bias of digital literacy on household green cooking energy consumption caused by missing variables. The level of individual characteristics includes factors such as the age of the household head, the square of age, gender, education level, and health status. The level of family characteristics includes per capita household income, family size, total family assets, and family favor expenditures. The village level selects Village terrain. The definitions, assignments, and descriptive statistics of specific variables are shown in Table 1.

### 3.3. Variance Analysis of Variable Mean

Assessing the average difference between green cooking energy and non-green cooking energy is helpful to further analyze the impact of farmers’ digital literacy on farmers’ household green cooking energy. In this paper, the mean values of green cooking energy and non-green cooking energy are analyzed via *t*-test. The specific results are shown in Table 2. Specifically, among the green cooking energy users, the digital literacy of farmers is higher than that of non-green cooking energy users. In terms of individual characteristics, among the users of green cooking energy, the heads of households are younger, have higher education levels, and the male farmers are lower in age than the female farmers, but their health status is at a low level. In terms of family characteristics, the household population of green cooking energy users is relatively small, and the total household assets are relatively high. At the village characteristic level, the terrain where the green cooking energy users are located is more inclined to the plain area. The above summary reflects the basic situation of the difference in the mean value of each variable. However, digital literacy is a representation of individual ability, which may cause endogenous problems due to reverse causality and missing variables, resulting in biased research conclusions. Therefore, it is necessary to construct an instrumental variable in the following regression analysis to further explore the impact of digital literacy on the green cooking energy consumption of households.

### 3.4. Methods

In this study, “green cooking energy” belongs to a binary variable. According to the structural characteristics of this data, this paper uses the binary Probit model to study the impact of digital literacy on the consumption of green cooking energy in rural households. The specific model is as follows:(7)$$\mathrm{Prob}\;\left({\mathrm{Energe}}_{i}=1\right)=\Phi \left(\beta_{0}+\beta_{1}{\mathrm{Digital}}_{i}+\beta_{2}X_{i}+{\;\beta }_{5}\upsilon_{i}\right)$$

In the formula, ${\mathrm{Energe}}_{i}$ represents the green cooking energy consumption of the i household, ${\mathrm{Digital}}_{i}$ represents the digital literacy status of the i farmer, $X_{i}$ represents the control variables at the individual, household, and village levels, and $\upsilon_{i}$ is the random error term. $\beta_{1}$ is the influence of digital literacy on the green cooking energy consumption of rural households. A positive value means that the improvement of digital literacy is beneficial to rural households to strengthen their consumption of green cooking energy, and a negative value means that the improvement of digital literacy has an inhibitory effect on the green cooking energy consumption of rural households. At the same time, the robust standard error method is used to estimate the heteroscedasticity in the cross-section data.

## 4. Results

### 4.1. Benchmark Regression Analysis

In Table 3, the core explanatory variables are first put into the model via stepwise regression, and then individual characteristic variables, family characteristic variables, and village characteristic variables are sequentially put into the model, which can effectively reduce the error of regression results caused by missing variables. According to the regression results of model 1 to model 4, it can be found that the influence and significance of digital literacy on household green cooking energy consumption are stable, always significantly positive at the level of 1%, indicating that the higher the level of digital literacy of farmers, the higher the probability of farmers choosing green cooking energy consumption. Therefore, research hypothesis H1 is proved.

Among the control variables, the age of the head of household bears a significant positive impact on the green cooking energy consumption of the households. However, the age square is significantly negative at the level of 10%, indicating that the impact of age on the green cooking energy consumption of the households is inverted “U” type, i.e., with the increase in age, the households gradually change preference from green cooking energy to non-green cooking energy, which is consistent with the research conclusion of Twumasi et al. [16] in the credit availability and rural households’ clean energy consumption. When the gender of the head of the household is male, the head of household is more inclined to restrain the use of green cooking energy. The theory of planned behavior emphasizes that gender is an important factor that affects an individual’s behavior intention and specific behavior [19]. In traditional Chinese society, the family relationship is represented as “the male lead outside, while the female lead inside”. Therefore, as the main users of domestic energy in rural households, women are more willing to use efficient and convenient clean energy [12], while men have a negative impact on the use of green cooking energy in households. The education level of the head of household has a significant positive impact on the green cooking energy consumption of the households. The higher the education level, the greater the consideration given to the role of health effects in energy consumption choices, thus reducing the consumption of firewood and other traditional energy sources [11]. At the level of family characteristics, the per capita household income has a significant positive impact on the household green cooking energy consumption, indicating that with continuous growth in the income level of farmers, the demand for traditional renewable energy in rural areas continues to decrease, while the demand for new green energy continues to increase [43]. The household size has a significant negative impact on the household green cooking energy consumption. This may be due to the fact that the larger the household size, the higher the demand for cooking energy. Green cooking energy is mostly clean commercial energy. The higher the demand, the higher the energy consumption expenditure. Therefore, households are more inclined to use renewable-free biomass energy. On the other hand, the larger the household size, the higher the possibility of choosing to work abroad. The elderly left behind by the households become decision-makers, and therefore they are more inclined to consume traditional cooking energy. Total household assets have a significant positive impact on farmers’ adoption of green cooking energy, indicating that the greater the number of total household assets, the more likely they are to use green cooking energy. At the village characteristic level, the terrain has a significant positive impact on farmers’ household green cooking energy consumption. The flat terrain is conducive to the construction of energy infrastructure and can provide sufficient green energy supply for farmers.

### 4.2. Endogenous Test

Although the stepwise regression method is adopted in the benchmark regression, endogenous problems still remain a possibility due to other missing variables and reverse causality, which may lead to erroneous regression results. On the one hand, the adoption of green cooking energy by households may be affected by other factors, such as consumer preferences, cultural practices, government policies, and other unobservable variables; on the other hand, whether the households use green cooking energy has a potential impact on the individual digital literacy of the households, e.g., the household members who use green energy have higher education level and income conditions, and these households have complete conditions to support the improvement of individual digital literacy. To avoid the endogeneity problem as much as possible, this paper employs the existing research methods [44,45] for reference, and selects “the average value of digital literacy of other samples living in the same village except for the farmers themselves” as a tool variable to modify the above regression results. Given the similarity of digital literacy within the same village, individual digital literacy is affected by the average digital literacy of others within the same village. Meanwhile, the household consumption of green cooking energy is not directly related to other people’s digital literacy level. Theoretically, the selection of the above-mentioned tool variables meets the requirements of relevance and exogeneity. In this paper, the IV-Probit model and CMP model are used to analyze the energy consumption of household green cooking. The CMP model can effectively estimate the mixed model system with different explanatory variables, and the mixed model offers special advantages when there are categorical variables and censored data variables as endogenous variables [46].

Table 4 reports the regression results for instrumental variables. From the regression results in column (1) of Table 4, it can be seen that the Wald exogeneity test rejected the original assumption that there is no endogeneity in farmers’ digital literacy. This indicates that the results of instrumental variables are significantly different from the original results. After correcting the endogeneity, digital literacy is still significantly positive at the level of 1%, which further demonstrates that digital literacy has a significant positive impact on the green cooking energy consumption of households. Atanhrho_12 in column (2) represents the residual correlation of the two-stage regression model. According to the regression results in Table 4, its coefficient is significant at the level of 1%, indicating the existence of endogeneity among the models and it is necessary to use the joint model to test. Under the condition that the other variables are controlled, the estimation coefficient of digital literacy is significantly positive, and the regression coefficient is significantly higher than that of the benchmark model, which indicates that the positive impact of digital literacy on household green cooking energy consumption will be underestimated without endogenous treatment by using instrumental variables.

### 4.3. Robustness Test

In the benchmark regression, the measurement of farmers’ digital literacy is measured by the unified entropy method. However, this measurement method does not take into account the development of digital infrastructure between different provinces and the status of individual human capital. Thus, it is likely to produce measurement errors of digital literacy, which will lead to errors in the regression results. Therefore, this paper uses the replacement variable method and the replacement regression model to test the robustness. This paper adopts the research method of Asfaw et al. [47] for reference. First, it sums up the different structural literacy of each province and city in the sample separately. Second, it calculates the proportion of individual farmers’ literacy among different scenarios in different provinces and then carries out the equal-weighted assignment. Finally, it calculates the actual digital literacy of farmers. Furthermore, this paper directly sums up the scores of the five indicators to represent the individual farmers’ digital literacy level. In addition, the Probit model is changed into the Logit model. Moreover, the energy consumption intensity of green cooking in households is measured by the “monthly electricity charge” in the CFPS. According to the regression results listed in (1)–(2) in Table 5, whether the calculation method is changed or added up directly, digital literacy at the 1% level bears a positive impact on the energy consumption of green cooking in households. The regression results in column (3) show that digital literacy still has a significant impact after changing the regression model, which further demonstrates the reliability of the benchmark regression results, and indicates that improvements in digital literacy can effectively promote the consumption of green cooking energy by farmers. The regression results in column (4) show that improvements in digital literacy can effectively increase the consumption of electricity by households, which supports the impact of digital literacy on the consumption of green cooking energy by households.

### 4.4. Heterogeneity Analysis

The above article verified the promotion of digital literacy on green cooking energy consumption of farmers’ households, and then the heterogeneity analysis was carried out according to the regional development level and income level. The selection of regional development level as the dividing standard is due to: the unbalanced development between regions since the reform and opening up, advanced development in the eastern region and backward development in the western region [48], large differences in regional infrastructure construction and policy support, differences in development level among various regions in the cultivation of individual digital literacy, such as the construction and improvement of regional digital infrastructure, and differences in individual human capital. Choice of income was selected as another criterion because the energy accumulation theory indicates that with continuous income growth, the household energy consumption structure becomes increasingly optimized and gradually changes to clean energy.

This paper draws on existing research [49] to divide both the regional background and the samples according to the eastern, central, and western departments. According to the regression results listed in (1)–(3) in Table 6, it can be observed that digital literacy has a significant positive effect on the household green cooking energy consumption at 5% level in the eastern and western regions, while it has no significant effect in the central region. This may be due to an imbalance in regional economic development and differences in regional natural resources. Compared with the eastern and western regions, the demand levels of farmers in the central region are different, mainly due to the abundant clean energy supply in the western region and the perfect digital economic development in the eastern region, resulting in an insignificant effect of digital literacy in the central region. This fully demonstrates that the regional development level and the supply of clean energy have affected the promotion of digital literacy on green cooking energy consumption to a certain extent. At the same time, according to the average family income per capita as the dividing line, the sample is divided into two levels: high-income farmers and low-income farmers. The regression results listed in (4)–(5) in Table 6 show that the impact of digital literacy on green and clean energy consumption of households is not significant in high-income households, but significantly positive in low-income households, indicating that the promotion effect of digital literacy on green cooking energy consumption is more conspicuous in low-income households, while the difference in impact of digital literacy mainly lies in the transformation and upgrading of energy consumption structure of high-income households and insufficient green energy consumption of low-income households [50].

### 4.5. Mechanism Analysis

To verify whether digital literacy can increase the consumption of green cooking energy by affecting farmers’ non-farm employment, this paper will consider the impact of digital literacy on non-farm employment and non-farm employment stability. The stability of non-agricultural employment refers to the practice of the existing research work [51] and takes whether to sign a labor contract as the judgment standard. The regression results are shown in Table 7. It can be found that the coefficient of non-agricultural employment in column (1) is significantly positive, indicating that improvements in digital literacy significantly promote the probability of farmers’ participation in non-agricultural employment, thus contributing to an increase in total household income and providing a solid material basis for household green cooking energy consumption. The regression results in column (2) show that the coefficients of digital literacy and non-farm employment are still significantly positive, indicating that the digital literacy of farmers promotes household green cooking energy consumption by promoting non-farm employment. The regression results of columns (3) and (4) show that digital literacy also promotes household green cooking energy consumption by improving the stability of farmers’ non-farm employment, thus proving the H2 hypothesis.

In CFPS2018, there are some questions about the importance of TV, Internet, newspapers, periodicals, and radio as information acquisition channels for the interviewed farmers (there are 1–5 grades, with 1 being very unimportant and 5 being very important). This paper draws on the research practice of Zhang [52], and after summarizing the above indicators, divides the new information channel indicators into five grades. The higher the grade, the more common it is for farmers to use digital technology channels to acquire information. The regression results of columns (5) and (6) show that digital literacy can promote the consumption of household green cooking energy by widening the information access channels of farmers, which fully shows that bridging the information gap of farmers can effectively promote the optimization and upgrading of household energy consumption structures, thus proving the H3 hypothesis.

## 5. Conclusions and Implications

### 5.1. Discussion

This paper uses theoretical and empirical analyses to verify the positive impact of digital literacy on the green cooking energy of farmers’ households. The findings presented herein are similar to those of He et al. [21] and Lange et al. [53], that is, digital literacy, as the deep application of digital technology and the enhancement of digital capabilities, can effectively promote the green cooking energy consumption of farmers’ households. The main contributions of this paper include: First, under the premise that the construction of digital villages drives the realization of rural revitalization and common prosperity, studying the impact of individual digital literacy among farmers on household green cooking energy consumption is of great significance in promoting the transformation of rural energy consumption, and thus enriches the existing literature on rural energy consumption; Second, the corresponding mechanism that digital literacy affects farmers’ household green cooking energy consumption is further analyzed, and expands the mechanism analysis from a multi-dimensional perspective; Third, the instrumental variable method is used to reduce the error of regression results caused by endogeneity. Of course, there are still many deficiencies in this research. First, this paper uses cross-sectional data and is thus unable to answer the time trajectory of digital literacy on green cooking energy consumption of households. Second, the sample data are from regions located within China, and the conclusions drawn may only apply to China or developing countries. Therefore, in the follow-up research, we need to construct panel data to analyze the long-term effect of digital literacy and employ international data for corresponding analysis.

### 5.2. Conclusions

This paper uses a sample of farmers from the CFPS2018 database to systematically explore the impact of digital literacy on the green cooking energy consumption of farmers’ households through the Probit model, and uses the mediating effect model to identify the impact mechanism of non-agricultural employment and information acquisition. The main research conclusions are as follows: (1) Improvements in digital literacy can effectively promote the consumption of green cooking energy in households. This conclusion remains valid after using the IV-Probit model and CMP model to solve the endogeneity; (2) The robustness test is conducted from three aspects: replacing the core explanatory variables, and changing the regression model and the explained variables. The results still show that digital literacy can significantly promote the household consumption of green cooking energy; (3) Heterogeneity analysis shows that the impact of digital literacy on the green cooking energy consumption of households varies among different regions and income levels, specifically in the eastern and western regions and low-income households. Improvements in digital literacy can significantly promote the green cooking energy consumption of households; however, among central region residents and high-income households the results were insignificant; (4) Mechanism analysis shows that digital literacy indirectly affects household green cooking energy consumption through its impact on non-agricultural employment and information access, revealing that the income effect and reductions in information asymmetry are conducive to promoting the transformation and upgrading of household cooking energy source.

### 5.3. Policy Recommendations

Based on the above conclusions, digital literacy can effectively promote the transformation and upgrading of household cooking energy consumption. This presents an important consideration of digital rural construction and the main dilemma of building a beautiful countryside to realize rural revitalization. To better play the positive role of digital literacy in promoting the ecological priority of low-carbon life’s development path, this paper proposes the following policy implications:(1)The construction of a digital countryside should not only consider the construction of digital infrastructure but also reasonably guide the cultivation of the internal digital literacy of the construction subject. First, it is necessary to support the construction of rural digital literacy education systems, enrich the supply of high-quality digital resources, explore the lifelong path of digital learning, strengthen farmers’ use of digital tools and digital software, consistently improve farmers’ digital literacy level, narrow the digital divide between urban and rural areas, and enhance farmers’ ability to occupy digital welfare; Second, it is necessary to improve farmers’ digital living standards, satisfy farmers’ sense of self-efficacy in digital technology, give full play to farmers’ initiative, actively participate in digital life, enhance their willingness to use digital tools, and jointly create a harmonious digital living atmosphere; Lastly, we should promote the further development of farmers’ digital skills, build a modern agricultural service system and a diversified training system, improve farmers’ ability to use digital “new farm tools”, and introduce social capital and non-profit organizations to participate in the improvement of farmers’ digital skills, so as to promote the extension of digital services and training to rural areas.(2)The cultivation of digital literacy should focus on both regional differences and key subjects, and implement accurate cultivation. Firstly, regional differences lead to heterogeneity in the improvement of individual digital literacy. Compared with the eastern and western regions, the central region should focus on giving play to the tangible hand of government policy regulation. In the construction of the “East-to-West Computing” project, it should actively undertake the transfer of relevant digital industries and part of the “computing power” role, cultivate the digital ecology in the central region from the top-level design and market, strengthen the cultivation of individual digital literacy, and at the same time play the spillover effect of digital agriculture to reverse promote improvements in individual digital literacy of farmers. Secondly, it is necessary to reasonably guide the digital training of different income groups. Compared with high-income groups, improvements in digital literacy can achieve a higher marginal effect. On the one hand, it is necessary to pay more attention to the digital cultivation of low-income groups, on the other hand, it is also necessary to strengthen the “digital–ecological” cultivation of high-income groups, and to guide and construct sustainable consumption concepts.(3)Give full play to the synergistic effect of digital literacy, and pay attention to farmers’ non-agricultural employment and information elements. Firstly, it is necessary to strengthen the orientation training of digital skills, meet the needs of the market, increase the opportunities for farmers to obtain non-agricultural employment, and prevent the decoupling of “demand and supply”. At the same time, it is necessary to increase support for new digital occupations, improve relevant policies and regulations, and provide policy support for farmers to develop diversified occupations. Secondly, it is necessary to broaden the information elements and build a diversified information circulation and access mechanism. Moreover, it is necessary to strengthen the construction of information security to reduce the additional losses of individual farmers caused by false information and to provide a security mechanism for farmers to enjoy digital welfare.

## Figures and Tables

**Table 1 ijerph-19-13464-t001:** Variable definitions and descriptive statistics.

Variable	Variable Definition and Assignment	Mean	Std. Dev
Green cooking energy	Non-green cooking energy = 0; Green cooking energy = 1	0.569	0.495
Digital literacy	Comprehensive calculation by entropy method	0.192	0.274
Age	Age of head of household: years old	48.35	10.87
Age squared	Age squared	2456	995.7
Gender	Female = 0; Male = 1	0.547	0.498
Education level	Below high school = 0;Above high school = 1	0.064	0.245
Health status	Self-rating: 1–5 (very healthy-unhealthy)	3.133	1.274
Per capita household income	Household income per capita (logarithm)	9.349	0.965
Family size	Family size (person)	4.095	1.971
Total household assets	Total household assets (logarithm)	11.22	3.426
Favor spending	Favor spending (logarithm)	7.255	2.334
Terrain	Non-plain = 0; Plain = 1	0.396	0.489

**Table 2 ijerph-19-13464-t002:** Based on the mean difference analysis of green cooking energy and non-green cooking energy.

Variable	Non-Green Cooking Energy	Mean	Green Cooking Energy	Mean	Mean Diff
Digital literacy	1768	0.172	2332	0.208	−0.036 ***
Age	1768	49.10	2332	47.78	1.320 ***
Age squared	1768	2527	2332	2401	126.048 ***
Gender	1768	0.570	2332	0.530	0.039 **
Education level	1768	0.037	2332	0.084	−0.048 ***
Health status	1768	3.201	2332	3.081	0.119 ***
Per capita household income	1768	9.089	2332	9.545	−0.457 ***
Family size	1768	4.143	2332	4.058	0.0840
Total household assets	1768	10.93	2332	11.45	−0.521 ***
Favor spending	1768	7.225	2332	7.278	−0.053
Terrain	1768	0.286	2332	0.479	−0.193 ***

Note: *** represent significance at 1% levels, ** represent significance at 5% levels.

**Table 3 ijerph-19-13464-t003:** Benchmark regression.

	(1)	(2)	(3)	(4)
Variable	Model 1	Model 2	Model 3	Model 4
Digital literacy	0.463 ***	0.396 ***	0.313 ***	0.302 ***
	(0.077)	(0.081)	(0.081)	(0.081)
Age		0.032 **	0.026 *	0.027 *
		(0.015)	(0.015)	(0.015)
Age squared		−0.000 **	−0.000 *	−0.000 *
		(0.000)	(0.000)	(0.000)
Gender		−0.096 **	−0.131 ***	−0.123 ***
		(0.043)	(0.044)	(0.044)
Education level		0.587 ***	0.477 ***	0.489 ***
		(0.103)	(0.106)	(0.106)
Health status		−0.031 *	−0.018	−0.015
		(0.017)	(0.017)	(0.017)
Per capita household income			0.280 ***	0.274 ***
			(0.039)	(0.038)
Family size			−0.018	−0.021 *
			(0.012)	(0.012)
Total household assets			0.014 **	0.013 **
			(0.007)	(0.007)
Favor spending			0.002	0.001
			(0.010)	(0.010)
Terrain				0.428 ***
				(0.054)
Regional effect	Y	Y	Y	Y
Constant	1.627 ***	1.105 *	−1.602 **	−1.966 ***
	(0.463)	(0.579)	(0.710)	(0.711)
Pseudo R2	0.108	0.118	0.149	0.159
Observations	4078	4078	4078	4078

Note: *** *p* < 0.01, ** *p* < 0.05, * *p* < 0.1, Robust standard errors in parentheses.

**Table 4 ijerph-19-13464-t004:** Endogenous test.

	(1)	(2)
Variable	IV-Probit	CMP
Digital literacy	3.972 ***	2.903 ***
	(1.140)	(0.410)
Control variable	Y	Y
Regional effect	Y	Y
Wald test	15.44	-
*p* value	0.000	-
atanhrho_12	-	−0.847 ***
		(0.213)
Constant	−1.556 **	−1.115 *
	(0.733)	(0.584)
Observations	4078	4100

Note: *** *p* < 0.01, ** *p* < 0.05, * *p* < 0.1, Robust standard errors in parentheses.

**Table 5 ijerph-19-13464-t005:** Robustness test.

	(1)	(2)	(3)	(4)
Variable	Model 1	Model 2	Model 3	Model 4
Digital literacy	0.010 ***	0.123 ***	0.495 ***	11.627 *
	(0.003)	(0.038)	(0.135)	(6.181)
Control variable	Y	Y	Y	Y
Regional effect	Y	Y	Y	Y
Constant	−2.019 ***	−2.342 ***	−3.711 ***	−95.335 **
	(0.711)	(0.729)	(1.326)	(45.793)
Pseudo R2	0.160	0.159	0.161	-
Observations	4078	4078	4078	4100

Note: *** *p* < 0.01, ** *p* < 0.05, * *p* < 0.1, Robust standard errors in parentheses.

**Table 6 ijerph-19-13464-t006:** Heterogeneity analysis.

	(1)	(2)	(3)	(4)	(5)
Variable	East	Central	West	High-Income Farmers	Low-Income Farmers
Digital literacy	0.309 **	0.256	0.305 **	0.111	0.365 ***
	(0.139)	(0.158)	(0.133)	(0.161)	(0.096)
Control variable	Y	Y	Y	Y	Y
Regional effect	Y	Y	Y	Y	Y
Constant	−3.253 ***	−2.326 **	−2.378 ***	−6.248 ***	−2.443 ***
	(0.961)	(0.939)	(0.839)	(1.284)	(0.613)
Pseudo R2	0.177	0.128	0.119	0.168	0.132
Observations	1469	1114	1495	1318	2750

Note: *** *p* < 0.01, ** *p* < 0.05, Robust standard errors in parentheses.

**Table 7 ijerph-19-13464-t007:** Intermediary test of digital literacy and household green cooking energy consumption.

	(1)	(2)	(3)	(4)	(5)	(6)
Variable	Non-Farm Employment	Green Cooking Energy	Non-Farm Employment Stability	Green Cooking Energy	Access to Information	Green Cooking Energy
Digital literacy	0.317 ***	0.277 ***	0.508 ***	0.282 ***	0.724 ***	0.272 ***
	(0.102)	(0.082)	(0.111)	(0.081)	(0.085)	(0.082)
Non-farm employment		0.661 ***				
		(0.090)				
Non-farm employment stability				0.366 ***		
				(0.104)		
Access to information						0.041 ***
						(0.016)
Control variable	Y	Y	Y	Y	Y	Y
Regional effect	Y	Y	Y	Y	Y	Y
Constant	−5.953 ***	−1.837 **	−4.525***	−1.922 ***	2.716 ***	−2.094 ***
	(0.784)	(0.734)	(1.156)	(0.706)	(0.493)	(0.707)
Pseudo R2	0.102	0.170	0.191	0.162	-	0.161
Observations	4100	4078	4100	4078	4100	4078

Note: *** *p* < 0.01, ** *p* < 0.05, Robust standard errors in parentheses.

## Data Availability

If necessary, we can provide raw data.

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
