# Peer review of "Research on the Impact of Digital Literacy on Farmer Households’ Green Cooking Energy Consumption: Evidence from Rural China"

_ijerph, 2022, doi:10.3390/ijerph192013464_

Round 1

Reviewer 1 Report

This paper attempts to study the impact of individual digital literacy on household green cooking energy consumption. In general, the research perspective has a certain novelty and the structure is basically reasonable, which can enriching the related achievements of the existing research results of rural energy consumption, and expand the mechanism analysis. However, there still exist some major drawbacks that need to be further addressed.

1. First of all, the definition of green cooking energy is crucial to this study. I think it is too arbitrary to classify the liquefied gas, natural gas, solar energy, and electricity as green cooking energy in this paper. For electricity, wind and hydropower could be defined as green cooking energy, while it is unreasonable to define thermal power as green cooking energy. This shortcoming may cause wrong research results.

2. The originality, In the introduction section, the research gap should be well justified, while this paper provide more information about the previous works by listing "who does what", the summary of the existing work is insufficient.

3. The discussion is not enough, neglecting the comparison with previous works, what are your new findings compared with existing works?

4. The results show that digital literacy is significantly positive for the green cooking energy consumption of households in the eastern and western regions, while it does not have a significant impact in the central region. Authors did not give sufficient explain of this result.

5. Half of the references cited in this paper are not cited from international journal. It is necessary to make a label for the non-international references.

Author Response

  1. First of all, the definition of green cooking energy is crucial to this study. I think it is too arbitrary to classify the liquefied gas, natural gas, solar energy, and electricity as green cooking energy in this paper. For electricity, wind and hydropower could be defined as green cooking energy, while it is unreasonable to define thermal power as green cooking energy. This shortcoming may cause wrong research results.

Thank you very much for your valuable comments! We note the confusion over the definition of green cooking energy in rural China, especially in the use of electricity. We fully agree that thermal power generation is an unclean energy production behavior, which will produce a large amount of harmful gases to pollute the air and the natural environment. However, it still needs to be pointed out that thermal power generation, as a production behavior of the energy supply end, can't completely transfer the pollution behavior to the consumption end. The individual's choice of electric energy consumption represents that he is more inclined to the efficient, convenient and clean attributes of electric energy. It is not the individual's choice that directly causes the environmental pollution caused by thermal power generation, especially in the vast rural society of China. If there is a lack of thermal power in cooking energy or living energy, farmers will generally choose biomass energy such as firewood and coal, thus directly causing a large area of environmental pollution. Therefore, in this paper, we define the energy that farmers do not directly produce a lot of harmful gases as green cooking energy (mainly including liquefied gas, natural gas, solar energy and electricity) as long as they use it in cooking activities, and at the same time, we add relevant literature for corresponding support. However, it is a very interesting field for the difference of thermal power generation, wind power generation and hydropower generation, and we will further explore it in the future research! The references are as follows:

Liu P, Han C, Teng M. Does clean cooking energy improve mental health? Evidence from China[J]. Energy Policy, 2022, 166: 113011.

He J, Qing C, Guo S, et al. Promoting rural households' energy use for cooking: Using Internet[J]. Technological Forecasting and Social Change, 2022, 184: 121971.

  1. The originality, In the introduction section, the research gap should be well justified, while this paper provide more information about the previous works by listing "who does what", the summary of the existing work is insufficient.

Based on the opinions of the review experts, we summarized the existing research work from the perspective of education, gender, age and income from the influencing factors of farmers' clean energy consumption. We also increased the literature discussion on the impact of non-agricultural employment perspective, subjective well-being, internet use and ecological public welfare perspective on farmers' clean energy use, and pointed out the shortcomings of the existing research.

  1. The discussion is not enough, neglecting the comparison with previous works, what are your new findings compared with existing works?

We have added a separate section to the conclusions and recommendations to make the structure of this article more perfect. In the discussion, the focus is to connect the existing research results with this paper, and further elaborate some limitations in this paper and the following research programs.

  1. The results show that digital literacy is significantly positive for the green cooking energy consumption of households in the eastern and western regions, while it does not have a significant impact in the central region. Authors did not give sufficient explain of this result.

In the heterogeneity analysis, we added the reason analysis that had no significant impact on the digital literacy in the central region. Digital literacy has a significant positive effect on the household green cooking energy consumption at 5% level in the eastern and western regions, while it has no significant effect in the central region. This may be due to the imbalance in regional economic development and the difference in regional natural resources. Compared with the eastern and western regions, the demand levels of farmers in the central region are different, and this difference is mainly due to the abundant clean energy supply in the western region and the perfect digital economic development in the eastern region, which results in the insignificant effect of digital literacy in the central region. This fully shows that the regional development level and the supply of clean energy have affected the promotion effect of digital literacy on green cooking energy consumption to a certain extent.

  1. Half of the references cited in this paper are not cited from international journal. It is necessary to make a label for the non-international references.

Non-international references will be noted in our references.

Reviewer 2 Report

The subject addressed in this paper is of particular importance, because the era of digital literacy is experiencing substantial growth. Digital literacy is the ability to use "new media", which provides the opportunity to actively participate in an increasingly digitized society.

The paper constructs a digital literacy index by the entropy method and uses a Probit regression model. As a first conclusion, improving digital literacy can effectively promote green energy consumption for cooking in households. Moreover, this conclusion is also valid after using the IV-Probit and the CMP model to solve the endogeneity. The heterogeneity analysis shows that the specific performance is in the eastern and western areas where the level of income is low. This is not the case in central or higher income regions. My question is what is the explanation of this phenomenon? Perhaps, it would be useful to better explain this aspect.

The conclusions of the paper are pertinent, that is, almost all the measures that could lead to an improvement of digital literacy, in the economic environment, in the political environment, are included.

Author Response

Thank you very much for your comments! The reasons for this phenomenon are explained as follows: (1) Digital literacy has a significant positive effect on the household green cooking energy consumption at 5% level in the eastern and western regions, while it has no significant effect in the central region. This may be due to the imbalance in regional economic development and the difference in regional natural resources. Compared with the eastern and western regions, the demand levels of farmers in the central region are different, and this difference is mainly due to the abundant clean energy supply in the western region and the perfect digital economic development in the eastern region, which results in the insignificant effect of digital literacy in the central region. This fully shows that the regional development level and the supply of clean energy have affected the promotion effect of digital literacy on green cooking energy consumption to a certain extent. (2) The impact of digital literacy on green and clean energy consumption of households is not significant in high-income households, but significantly positive in low-income households, indicating that the promotion effect of digital literacy on green cooking energy consumption is more obvious in low-income households, while the difference of the impact of digital literacy mainly lies in the transformation and upgrading of energy consumption structure of high-income households and insufficient green energy consumption of low-income households .

Reviewer 3 Report

The paper constructs a digital literacy index by entropy method and uses the Probit model regression and mediation effect model test to study the impact effect and theoretical mechanism of individual digital literacy on household green cooking energy consumption. The research results show that the improvement of digital literacy can effectively promote the consumption of green cooking energy in households. The heterogeneity analysis shows that the impact of digital literacy on green cooking energy consumption of households in different regions and different income stages is different. The specific performance is in the eastern and western regions and households in the low-income stage. Mechanism analysis shows that digital literacy has a significant positive impact on household green cooking energy consumption through non-agricultural employment and information acquisition. The cultivation of digital literacy should focus on key subjects, and implement precise cultivation. We should give full play to the synergistic effect of digital literacy, and pay attention to the non-agricultural employment of farmers and information elements.

This is a very interesting paper and I propose to publish it.

Author Response

Thank you very much for the accreditation of the review experts. Your accreditation is the driving force for our progress in academic research!

Reviewer 4 Report

The paper touches on an interesting topic. I am glad that I could read it and I thank the authors for their work.

The paper itself is properly structured and contains all the elements. This is an article that focuses heavily on the research part. The verification of research hypotheses was proposed.

The paper is aesthetic and the language used is adequate to the presented issue.

The paper seems to be very well written. The only issues that the authors could pay attention to and improve them are listed below:

- the title of Table 7 is not in English;

- the presented references are topical and timely. However, I believe that 40 literature items are definitely not enough. I propose to deepen the analysis of the literature and increase the number of references.

Author Response

We are very grateful to the review experts for controlling the details of this article and making our article more rigorous. The title of Table 7 has been corrected accordingly. At the same time, in the introduction, we added related literatures to the literature summary, which made up for the shortage of existing literatures.

Round 2

Reviewer 1 Report

All the deficiencies mentioned in the first round have been improved in the revised version.